# Synthesis and Biological Evaluation of Sclareolide-Indole Conjugates and Their Derivatives

**DOI:** 10.3390/molecules28041737

**Published:** 2023-02-11

**Authors:** Ying Cheng, Xilin Lyu, Chen Liu, Xiancheng Wang, Jing Cheng, Daizhou Zhang, Xiangjing Meng, Yujun Zhao

**Affiliations:** 1School of Chinese Materia Medica, Nanjing University of Chinese Medicine; Nanjing 210023, China; 2State Key Laboratory of Drug Research and Small-Molecule Drug Research Center, Shanghai Institute of Materia Medica, Chinese Academy of Sciences, 555 Zuchongzhi Rd., Shanghai 201203, China; 3University of Chinese Academy of Sciences, No.19A Yuquan Road, Beijing 100049, China; 4Shandong Provincial Key Laboratory of Biopharmaceuticals, Shandong Academy of Pharmaceutical Sciences, Jinan 250101, China; 5School of Pharmaceutical Sciences, Zhengzhou University, Zhengzhou 450001, China

**Keywords:** sclareolide, indole, polyphenol, acetal, antitumor activity

## Abstract

Sclareolide is a sesquiterpene lactone isolated from various plant sources in tons every year and is commercially used as a flavor ingredient in the cosmetic and food industries. Antitumor and antiviral activities of sclareolide have been previously reported. However, biological studies of sclareolide synthetic analogous are few. In view of these, we developed a robust synthetic method that allows the assembly of 36 novel sclareolide-indole conjugates and their derivatives. The synthetic method was based on TiCl_4_-promoted nucleophilic substitution of sclareolide-derived hemiacetal **4**, while electron-rich aryles including indoles, polyphenol ethers, and pyrazolo [1,5-a]pyridine were good substrates. The stereochemistry of the final products was confirmed by single-crystal X-ray diffraction analysis, while the antiproliferative activities of selected final products were tested in K562 and MV4-11 cancer cell lines. Cytometric flow analysis shows that lead compounds **8k**- and **10**-induced robust apoptosis in MV4-11 cancer cells, while they exhibited weak impact on cell cycle progression. Taken together, our study suggests that sclareolide could be a good template and substrate for the synthesis of novel antiproliferative compounds.

## 1. Introduction

Sclareolide (Figure 1) is a sesquiterpene lactone isolated from various plant sources including *Salvia sclarea*, *Salvia yosgadensis*, and tobacco leaves, among others [1,2]. It is commercially used as a flavor ingredient in the cosmetic [3,4] and food industries [2,5,6]. In 2016, the consumption volume of sclareolide was up to 100 metric tons in European countries, which represents 82% of its global consumption volume [7]. It has also been reported that sclareolide exhibited robust antitumor activity [8] and anti-viral activity [9]. For example, sclareolide significantly improved the antitumor activities of Gemcitabine, an FDA approved anticancer drug, against pancreatic cancer cells in vitro and in vivo [8]. In addition, sclareolide blocked the Ebola virus fusion process with an inhibitory EC_50_ value of 8.0 μM and exhibited wide-spectrum activities against additional seven filoviruses [9]. These interesting biological activities of sclareolide have motivated an increasing number of studies in this area.

In natural product synthesis, sclareolide is often used as a starting material for the synthesis of many terpenoids and their derivatives [7,10,11,12,13,14], while a few reports have also documented synthesis of sclareolide analogues and the examination of their biological activities [15,16,17,18,19,20,21]. For instance, González and coauthors reported the synthesis of sclareolide derivatives as well as the evaluation of their cytotoxic, antifungal, and antiviral activities in vitro [18]. Recently, Tran et. al. reported syntheses of 72 analogues of the natural product andrographolide using sclareolide as a key starting material [17]. One lead compound from this series was a potent inhibitor against two low-passage Chikungunya virus isolates from human patients. Despite these recent achievements, to the best of our knowledge, neither sclareolide-indole conjugate nor sclareolide-polyphenol conjugate has been reported, as shown in Figure 1. In view of the interesting biological activities of sclareolide and its derivatives, we aimed to synthesize novel sclareolide derivatives containing various indoles or polyphenol methyl ethers, given their high prevalence in bioactive compounds and drugs [19,22,23,24,25,26,27,28]. As part of our ongoing interest [29] in finding natural product derivatives with potent antitumor activities, we herein report our synthesis and the biological evaluation of a new series of sclareolide-indole conjugates and their derivatives.

## 2. Results and Discussion

Our synthesis started with NBS-promoted bromination of commercially available sclareolide (**1**, Figure 1), while a bromination compound **2** was obtained in a 53% yield. It is very interesting that bromination only took place on β-position of the ester group. In the presence of lithium hydroxide, a bromine atom of **2** was eliminated and a previously known α, β-unsaturated lactone **3** [30,31,32,33] was obtained in an 85% yield, without observation of hydrolysis of the ester group. The stereochemistry of **3** was further verified by single-crystal X-ray diffraction analysis (Figure 2) [34]. Having confirmed the structure of **3**, subsequent reduction in the presence of DIBAL-H yielded hemiacetal **4** in 98% yield as a mixture of isomers. It is worth of noting that >1 g of **4** could be obtained in a single run of reaction, which makes it feasible to synthesize its diverse derivatives. 

Previously, one of us was involved in studies of Lewis acid-chiral acetal-promoted polyene cyclization [35,36,37] in which oxocarbenium generated from an acetal could be attacked by a weak nucleophile, an olefin [38]. Therefore, we examined whether a hemiacetal **4** could react with an electron-rich indole **5** in the presence of Lewis acid, and the results are summarized in Table 1. TiCl_4_ was found to be the acid of choice and expected product **6** was obtained in 57% yield (Entry 6), while BF_3_-OEt_2_, InBr_3_, In(OTf)_3_, and SnCl_4_ yielded a low yield of or no **6** (entries 1–5). The isolated yield of **6** slightly decreased when 1.5 equiv. of **5** was used (entry 7). Interestingly, the highest yield of **6** in 59% with an isomer ratio of 69:31 was obtained when 0.8 equiv. of TiCl_4_ was used (entry 9).

With an optimized reaction condition in hand, indoles bearing various substituents were examined to explore substrate scope and the results were summarized in Table 2. Overall, thirty substituted indoles were tested, seven desired products (**8b**, **8h**, **8m**, **8v**, **8z**, **8ab**, and **8ad**) were obtained in >80% yields, and twenty-six desired products were furnished in >50% yield. In general, the 4, 5, and 6 positions of indole tolerate electron-withdrawing and electron-donating groups as exemplified in **7a** to **7y**, resulting **8a**–**8y** with a moderate to good yield. In addition, the isomer ratios of **8a**–**8y** were dynamic in a range of 51:49 to 85:15. Surprisingly, **7z** bearing a larger pinacolboranyl group (-BPin) at the 4-position provided no desired product, suggesting the steric hindrance was detrimental to the reaction yield. Similarly, indole **7af** bearing a large butyl group at the 2-position also yielded the desired product **8af** in a relative low yield (35%) with an isomer ratio of 87:13, which suggests that steric hindrance was detrimental for the reaction yield. Nevertheless, **7ab**–**7ae** containing 2-methyl, 2-ethyl, 2-phenyl, and 2-CO_2_Me groups all furnished the desired products in good yields with isomer ratios around 80:20. The stereochemistry of the major isomer of **8ab** was further confirmed by single-crystal X-ray diffraction analysis (Figure 3) [34] and an ^1^H NMR study of signal crystals. Overall, the sclareolide-indole conjugates **8a**–**8y** and **8ab**–**8ae** were readily obtained in moderate to good yields as a mixture of two isomers, whereas the steric-demanding **8z** was an outlier without observation of a desired product.

In view of the robust reactivity of electron-rich indoles in TiCl_4_ mediated coupling to **4**, other electron-rich aromatic rings were further examined and the results are summarized in Figure 2. Gratifyingly, polyphenol methyl ethers are good substrates and expected conjugation products **9–13** were obtained in good yields with isomer ratios ranging from 66:34 to 87:13.

In addition, pyrazolo [1,5-a]pyridine was a good substrate and N-heterocycle-containing product **14** was obtained in an 82% yield with an isomer ratio of 66:34. Free polyphenol phloroglucinol was also tested but no desired product was obtained, which is in contrast to phloroglucinol trimethyl ether affording **10** in a 62% yield. These results suggest that multiple free OH groups are not tolerated in the reaction condition. Other aromatic substrates, including anisole, veratrole, acetaminophen, and 2-pyridone, were also tested in the optimized reaction condition. Unfortunately, no expected coupling product was obtained. The limitations of the substrate scope were summarized in Appendix A.

Next, we examined the antiproliferative activities of selected sclareolide–indole conjugates and their analogues in K562 and MV4-11 cancer cell lines (Table 3). K562 is a human chronic myelogenous leukemia (CML) cell line and MV4-11 is a human acute myeloid leukemia (AML) cell line, both of which are considered sensitive blood cancer cell lines and are often used as screening cell lines for anticancer agents. [29,39,40] Decitabine, an FDA approved drug for the treatment of blood or bone marrow cancer, was used as a positive control, while sclareolide and **3** were also included as reference compounds. The antiproliferative activities results are summarized in Table 3. Overall, **8c**, **8g**, **8k**, **8m**, **8n**, **8p**, **8q**, **8s**, **8t**, **8w**, **8ab**, **8ad**, and **8ae** have inhibitory IC_50_ values in a range of 12.0 μM to 4.4 μM in K562 cells and in a range of 0.8 μM to 7.9 μM in MV4-11 cells. The best compound of this series was **8k**, which inhibited cell growth of K562 and MV4-11 cells with IC_50_ values of 5.2 ± 0.6 μM and 0.8 ± 0.6 μM, respectively, and **8k** is more potent than sclareolide that has IC_50_ values of 10.8 ± 0.6μM and 4.5 ± 0.3μM, respectively (Figure 4). For sclareolide–polyphenolic conjugates **9**, **10**, **11**, **12**, and **13**, their antiproliferative activity was very similar to that of sclareolide, while the best compound **10** was about two-fold more potent in K562 and MV4-11 cells. Last but not least, the antiproliferative activities of **14** and sclareolide have minimal differences.

In order to further elucidate the antiproliferative activities of **8k** and **10**, cytometric flow analysis was performed in MV4-11 to examine cell cycle arrest and apoptosis induction (Figure 5 and Figure 6). MV4-11 cells were treated with **8k** or **10** at the indicated concentrations for 48 h before the addition of dyes and subsequent sorting, while Decitabine was used as a positive control. The corresponding cells were counted based on their types, while their numbers were normalized and plotted, as shown in Figure 5 and Figure 6.

The data in Figure 5 shows that both **8k** and **10** dose-dependently arrest cell cycle progression at the S phase, while Decitabine has no impact on cell cycle progression at the concentrations tested. The percentage of the S phase for DMSO, all Decitabine-treated groups, and the 0.3μM **8k**/**10**-treated groups is 42%, while that of the 10 μM **8k**/**10**-treated groups is 45.8 ± 1.3% and 45.9 ± 1.9%, respectively, representing a small amount of increase of 3%. These data indicate that **8k** and **10** have weak effects on the cell cyle progression of MV4-11 cells up to 10 μM.

The data in Figure 6 show that both **8k** and **10** dose-dependently-induced cell apoptosis, while Decitabine was also an effective apoptosis inducer at the concentration tested. In the DMSO-treated group, the percentage of apoptotic cells was 3.89 ± 0.12%, while that in all Decitabine treated groups were in a range of 8.66% to 11.4%, representing a 2~3 fold increase compared with DMSO; **8k** and **10** were stronger apoptosis inducers at 30 μM compared with Decitabine, having apoptotic cells at 18.3 ± 0.8% and 12.3 ± 0.3%, respectively. These data suggest that **8k** and **10** have strong effects on the apoptotic cell death of MV4-11 cells, which partially explains their antiproliferative phenotype in cell growth inhibition assays.

## 3. Materials and Methods

### 3.1. General Chemistry Methods

All reactions were conducted in a round-bottomed flask equipped with a Teflon^®^-coated magnet stirring bar. Experiments involving moisture and/or air sensitive components were performed under a nitrogen atmosphere. Commercial reagents and anhydrous solvents were used without further purification. The crude reaction products were purified by flash column chromatography packed with silica gel. Proton nuclear magnetic resonance (^1^H NMR) was performed in Bruker Advance 400 NMR spectrometers (Bruker corp., Billerica, MA, USA). Carbon nuclear magnetic resonance (^13^C NMR) spectroscopy was performed in Bruker Advance 500 NMR spectrometers (Bruker corp., Billerica, MA, USA). High resolution ESI mass spectrum analysis was performed on Agilent Q-TOF mass spectrometer (G6520, Agilent Technologies Inc., Santa Clara, USA). The analytical UPLC model was Waters Acquity H class (UV detection at 230 nm and 254 nm, Waters Corp., Milford, USA) and the reverse phase column used was the Acquity UPLC^®^ BEH (C18–1.7 µm, 2.1 × 50 mm, Waters Corp., Milford, USA). Further purification of final compounds for biological testing was performed on a preparative HPLC (Waters 2545, Waters Corp., Milford, USA) with a C18 reverse phase column (Waters Corp., Milford, USA). The mobile phase used here was a gradient flow of solvent A (water) and solvent B (CH_3_CN) at a flow rate of 10 mL/min. All final compounds were purified to ≥95% purity as determined by analytical UPLC analysis. Isomer ratio was determined by integration of corresponding vinyl proton or O-CH peaks in ^1^H NMR spectrum.

### 3.2. Procedures for Synthesis of Sclareolide Derivatives

Synthesis of (3aR,5aS,9aS,9bS)-9b-bromo-3a,6,6,9a-tetramethyldecahydronaphtho [2,1-b]furan-2(1H)-one (**2**): A 250 mL round-bottom flask equipped with a magnetic stirring bar was charged with **1** (5.0 g, 19.97 mmol, 1.0 equiv), N-Bromosuccinimide (8.89 g, 49.92 mmol, 2.5 equiv), dibenzoyl peroxide (483.7 mg, 1.20 mmol, 0.1 equiv), and CCl_4_ (60 mL). The mixture was stirred at 85 °C for 16 h. The resulting mixture was filtered, and the filtrate was concentrated. The crude was purified by flash column chromatography to afford the title compound **2** as a white solid (3.48 g, 53%). ^1^H NMR (500 MHz, CDCl_3_): δ 3.54–3.46 (d, J = 18.0 Hz, 1H), 3.14 (d, J = 18.0 Hz, 1H), 2.25–2.18 (m, 1H), 1.91 (dtd, J = 12.9, 3.3, 1.6 Hz, 1H), 1.75 (s, 3H), 1.71–1.61 (m, 2H), 1.57–1.43 (m, 3H), 1.43–1.32 (m, 2H), 1.29 (s, 3H), 1.19–1.10 (m, 2H), 0.96 (s, 3H), 0.89 (s, 3H). ^13^C NMR (126 MHz, CDCl_3_): δ 175.60, 94.65, 88.77, 49.55, 45.47, 43.87, 41.30, 38.50, 36.48, 33.82, 33.52, 26.18, 21.72, 18.88, 18.83, 17.98. HRMS (EI): m/z calculated for C_16_H_25_^79^BrO_2_ [M]^+^: 328.1038, Found: 328.1033. Specific Rotation: [α]^20^ = -46.95° (c = 1.0 g/100 mL, in CHCl_3_).

Synthesis of (3aR,5aS,9aS)-3a,6,6,9a-tetramethyl-4,5,5a,6,7,8,9,9a-octahydronaphtho [2,1-b]furan-2(3aH)-one (**3**): Dissolve **2** (3.48 g, 10.58 mmol, 1.0 equiv) in H_2_O and THF (20/20 mL) followed by addition of LiOH·H_2_O (3.55 g, 84.62 mmol, 8.0 equiv). The mixture was stirred at ambient temperature for 12 h. When the reaction was completed, the pH was adjusted to be 6 with 1N HCl. The aqueous layer was extracted with ethyl acetate (3 × 50 mL), and the combined organic layers were washed with brine (50 mL), dried over anhydrous Na_2_SO_4_, and concentrated in vacuo. The residual crude product was purified by flash column chromatography to afford the title compound **3** as a white solid (2.24 g, 85%). ^1^H NMR (500 MHz, CDCl_3_): δ 5.51 (s, 1H), 2.29 (dd, J = 8.9, 2.9 Hz, 1H), 1.83–1.68 (m, 3H), 1.63–1.57 (m, 1H), 1.56–1.53 (m, 4H), 1.51–1.50 (m, 1H), 1.49–1.44 (m, 2H), 1.22–1.15 (m, 4H), 0.96–0.93 (m, 1H), 0.90 (d, J = 7.8 Hz, 6H). ^13^C NMR (126 MHz, CDCl_3_): δ 197.98, 173.72, 107.09, 84.78, 57.61, 41.49, 40.71, 39.66, 37.09, 34.73, 33.24, 25.22, 22.36, 19.36, 18.07, 17.91. HRMS (EI): m/z calculated for C_16_H_24_O_2_ [M]^+^: 248.1776, Found: 248.1758. Specific Rotation: [α]^20^ = −157.13° (c = 1.0 g/100 mL, in CHCl_3_).

Synthesis of (3aR,5aS,9aS)-3a,6,6,9a-tetramethyl-2,3a,4,5,5a,6,7,8,9,9a-decahydro naphtho [2,1-b]furan-2-ol (**4**): A 100 mL dry round-bottom flask equipped with a magnetic stirring bar was charged with **3** (1.2 g, 4.83 mmol, 1.0 equiv) and anhydrous CH_2_Cl_2_ (40 mL). Subsequently, DIBAL-H (1.0 M in hexane, 8.7 mL, 8.7 mmol, 1.8 equiv.) was added at −78 °C. The resulting solution was allowed to stir from −78 °C to 0 °C for 12 h. The reaction mixture was quenched with water (30 mL) followed by addition of 2N NaOH (10 mL). The aqueous layer was extracted with CH_2_Cl_2_ (3 × 40 mL), and the combined organic layers were washed with brine (50 mL), dried over anhydrous Na_2_SO_4_, filtered, and concentrated in vacuo to afford the crude product **4** as a white solid (1.18 g, 98%). The crude product **4** was used for next step without further purification. ^1^H NMR (500 MHz, CDCl_3_): δ 5.91 (d, J = 1.5 Hz, 1H), 5.19 (d, J = 1.5 Hz, 1H), 2.10–1.95 (m, 1H), 1.72–1.55 (m, 4H), 1.49–1.44 (m, 4H), 1.40–1.34 (m, 2H), 1.15–1.03 (m, 6H), 0.82 (d, J = 2.0 Hz, 6H). ^13^C NMR (126 MHz, CDCl_3_): δ 161.79, 113.90, 102.55, 86.94, 55.52, 42.41, 42.05, 37.86, 37.65, 33.71, 33.47, 28.70, 21.58, 19.98, 18.99, 18.55.

### 3.3. Procedure for Coupling Reactions of Sclareolide-Indole and Sclareolide-Aromatic Compounds **6**, **8a**-**8y**, **8ab**-**8ae**, and **9**–**14**

Synthesis of 3-((3aR,5aS,9aS)-3a,6,6,9a-tetramethyl-2,3a,4,5,5a,6,7,8,9,9a-decahydro naphtha [2,1-b]furan-2-yl)-1H-indole (**6**): To a 100 mL dry round-bottom flask equipped with a magnetic stirring bar, **4** (0.20 mmol, 1.0 equiv), indole (0.16 mmol, 0.8 equiv), and anhydrous CH_2_Cl_2_ (10 mL) were added. The solution was cooled to −78 °C then TiCl_4_ (1.0 M in CH_2_Cl_2_, 0.16 mmol, 0.8 equiv) was added dropwise via a syringe. The reaction mixture was stirred at −78 °C for 2.5 h before quenching with NaHCO_3_ saturated aqueous solution (15 mL). The aqueous layer was extracted with CH_2_Cl_2_ (3 × 30 mL), and the combined organic layers were washed with brine (50 mL), dried over anhydrous sodium sulfate, filtered, and concentrated in vacuo. The crude product was purified by flash column chromatography to afford the desired product **6** (white solid, 20.4 mg, 58% yield, Isomer ratio: 69:31). ^1^H NMR (500 MHz, CDCl_3_): δ 8.07 (s, 1H), 7.74 (ddd, J = 8.0, 5.6, 1.0 Hz, 1H), 7.35 (dt, J = 8.2, 1.0 Hz, 1H), 7.21–7.16 (m, 2H), 7.14–7.07 (m, 1H), 6.05 (d, J = 1.3 Hz, 0.69H), 6.02 (d, J = 1.3Hz, 0.31H), 5.50(d, J = 1.2Hz, 0.31H), 5.40 (d, J = 1.2 Hz, 0.69H), 2.07 (ddt, J = 12.7, 6.6, 2.8 Hz, 1H), 1.81–1.64 (m, 4H), 1.55–1.38 (m, 7H), 1.34–1.14 (m, 5H), 1.08 (ddd, J = 50.3, 12.2, 2.3 Hz, 1H), 0.94–0.88 (m, 6H). ^13^C NMR (126 MHz, CDCl_3_): δ 157.03, 156.15, 136.97, 136.78, 126.21, 126.09, 123.30, 122.81, 121.93, 119.74, 119.61, 119.45, 119.42, 118.09, 117.42, 117.05, 115.77, 111.47, 111.39, 87.67, 87.20, 78.79, 78.30, 56.06, 55.22, 53.55, 42.79, 42.33, 42.28, 42.20, 38.21, 38.18, 38.14, 37.87, 33.83, 33.77, 33.63, 33.54, 28.68, 26.63, 21.63, 21.58, 20.52, 20.33, 20.15, 19.80, 18.74, 18.68. HRMS (ESI): m/z calculated for C_24_H_31_NO [M+H]^+^: 350.2406, Found [M+H]^+^: 350.2484.

The synthesis of **8a**–**8y**, **8ab**–**8af**, and **9**–**14** used the same protocol for the synthesis of **6**. The procedures of these syntheses and characterization of **8a**–**8y**, **8ab**–**8af**, and **9**–**14** were summarized in the Appendix A. 

### 3.4. Cell Growth Inhibition Assay

K562 and MV4-11 cancer cell lines were purchased from National Collection of Authenticated Cell Cultures (Shanghai, China). STR analysis was performed and passed for all cell lines after in-house cell culture. The testing compounds were serially diluted in 96-well plates. Subsequently, the MV4-11 or K562 cells were seeded into wells at appropriate density (8000–25,000 cells/well). Cells in plates were incubated with 5% CO_2_ at 37 °C for 4 days. Thereafter, Cell Counting Kit-8 (Dojindo) was used to measure cell viability according to manufacturer’s protocols. The absorbance of OD450 were detected by a multimode microplate reader (TECAN SPARK 10M, Tecan Group Ltd., Männedorf, Switzerland). The untreated cells were set as an indicator of 100% cell viability. IC_50_ values were calculated by nonlinear regression analysis with GraphPad Prism 8 (Version 8.4.3).

### 3.5. Flow Cytometry Analysis of Cell Cycle Arrest

MV4-11 cells in logarithmic growth phase were seeded into 6-wells plates at a concentration of 1.5 × 10^6^ cells/mL and treated with **8k** and **10** at serial diluted concentrations for 48 h. Subsequently, all cells were collected, washed, and fixed in 2 mL 70% ethanol (precooled to −20 °C) at −20 °C for 12 h. The fixed cells were centrifuged, washed, and re-suspended in a cell cycle and apoptosis Kit solution (Beyotime Biotechnology, Code C1052). All samples were incubated at 37 °C in dark for 30 min prior to signal detection (FACSCalibur; BD Biosciences). Data were analyzed with FlowJo software (version 7.6.1) to provide to cell cycle progression status.

### 3.6. Flow Cytometry Analysis of Apoptosis

The MV4-11 cells in the logarithmic growth phase were seeded into 6-wells plates at a concentration of 1.5 × 10^6^ cells/mL, and treated with **8k** and **10** at the concentrations of 0.3, 1, 3, 10, 30, and 60 μM for 48 h, accordingly. After treatment, all the cells were collected and re-suspended cells in buffer at a concentration of 1.0 × 10^6^ cells/mL. One hundred μL of the solution (1.0 × 10^5^ cells) was transferred to a 5 mL EP tube, followed by addition of 5 μL of FITC Annexin V and 5 μL of Propidium Iodide (PI). The mixture of cells was gently vortexed and incubated for 15 min at room temperature in the dark. Four hundred μL of binding buffer was added to each EP tube prior to flow cytometry analysis. Propidium Iodide and AnnexinV staining were performed using FITC Annexin V Apoptosis Detection Kit I from BD Biosciences. Flow cytometry data were acquired on a FACS Calibur (BD).

## 4. Conclusions

In summary, we have synthesized a class of sclareolide–indole conjugates and their derivatives based on TiCl_4_-promoted nucleophilic substitution of hemiacetal **4**. Electron-rich aryles including indoles, polyphenol ethers, and pyrazolo [1,5-a]pyridine are good substrates for this reaction. The conjugation products were obtained in a moderate to good yield as a mixture of two isomers. The stereochemistry of the final products was confirmed by single-crystal X-ray diffraction analysis, while the antiproliferative activities of selected final products were tested in K562 and MV4-11 cancer cell lines. Cytometric flow analysis shows that lead compounds **8k**- and **10**-induced robust apoptosis in MV4-11 cancer cells, while they exhibited a weak impact on cell cycle progression. Our study suggests that sclareolide could be a good template and substrate for the synthesis of novel antiproliferative compounds.

## Data Availability

Not applicable.

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
