# Peer review of "Synthesis and Biological Evaluation of Sclareolide-Indole Conjugates and Their Derivatives"

_molecules, 2023, doi:10.3390/molecules28041737_

Round 1

Reviewer 1 Report

I congratulate the authors for their good work. I recommend a few changes in the manuscript.

1. Please separate the methodology and characterizations with two different headings.

2. What is CD or code in the main manuscript and supporting information file? Remove it from the main manuscript and supporting file. 

3. In figure. 4 on x-axis, the symbol is unidentified. What is that symbol?

4.  replace the colon with dot in all the captions

5.  N-heterocycle with N-heterocycle

Author Response

I congratulate the authors for their good work. I recommend a few changes in the manuscript.

Response: Thanks very much for careful review and constructive comments on our work.

  1. Please separate the methodology and characterizations with two different headings.

Response: Thanks for kind suggestion. As suggested, we have checked official Molecules Template and published papers of Molecules journal for a proper format. Unfortunately, we cannot find a proper way to include two different headings. Therefore, no additional heading was added.   

  1. What is CD or code in the main manuscript and supporting information file? Remove it from the main manuscript and supporting file. 

Response: Codes have been removed from the manuscript and supporting info except for Figures 5 and 6 where code numbers of 8k and 10 were included. We feel it will be easier to refer the most potent bioactive compounds as code numbers rather than 8k and 10 in the future, if they will be cited.

  1. In figure. 4, on x-axis, the symbol is unidentified. What is that symbol?

Response: the symbol of the x-axis is concentration of compounds in μM.

  1. Replace the colon with dot in all the captions

Response: the corrections have been done and thanks for suggestion.

  1. N-heterocycle with N-heterocycle

Response: the corrections have been done and thanks for suggestion.

Reviewer 2 Report

Hi. I thought this was a very nice paper with some synthesis. I think your bromination is OK because in the experimental you have two downfield protons with a coupling constant of 18 Hz. This I think must be two geminal protons deshielded by the electron withdrawing carbonyl group. There is also another tertiary hydrogen but it is more hindered. What I dont like is your crystal structure drawings which are not very good. Is that a poor data set and disorder in the lattice? Normally I can see stereochemistry straight away but with yours I had to look at the text drawings to see which groups were cis or trans. Anyhow I think whatever the data set quality you were able to resolve to a definite structure which is helpful. When you treated your precursor with TiCl4 Im surprised the ring methyl group does not migrate across to the coupling position giving a 2-methylfuran derivative. At least some of it. 

Author Response

Response: Thanks very much for careful review and constructive comments on our work.

(1) The crystal structure drawings were generated by the crystallographer and adopted with further modification. The data has passed qualification examination without apparent flaws. For more clear visualization of the structures, please download corresponding cif. files with CCDC access numbers 2236373 and 2238307 at webpage https://www.ccdc.cam.ac.uk/structures/? The information is free of charge.

(2) We did not observe methyl migration product. The methyl is beta position of cationic carbon center, while methyl migration more likely happens when it is alpha position to a cationic carbon center, such as pinacol rearrangement.